# Loneliness amongst Low-Socioeconomic Status Elderly Singaporeans and its Association with Perceptions of the Neighbourhood Environment

**DOI:** 10.3390/ijerph16060967

**Published:** 2019-03-18

**Authors:** Liang En Wee, Tammy Yun Ying Tsang, Huso Yi, Sue Anne Toh, Geok Ling Lee, Jaime Yee, Shannon Lee, Kellynn Oen, Gerald Choon Huat Koh

**Affiliations:** 1Department of Infectious Diseases, Singapore General Hospital, Singapore 169608, Singapore; 2Duke-NUS Graduate Medical School, Singapore 169857, Singapore; 3Institute of Mental Health, Singapore 539747, Singapore; tammy.tsang@mohh.com.sg; 4Saw Swee Hock School of Public Health, National University Health System, National University of Singapore, Singapore 119228, Singapore; ephyh@nus.edu.sg (H.Y.); ephkohch@nus.edu.sg (G.C.H.K.); 5Regional Health System Planning and Development, National University Health System, National University of Singapore, Singapore 119228, Singapore; mdcsates@nus.edu.sg; 6Department of Social Work, Faculty of Arts and Social Sciences, National University of Singapore, Singapore 119077, Singapore; swklgl@nus.edu.sg; 7Yong Loo Lin School of Medicine, National University Health System, National University of Singapore, Singapore 119228, Singapore; jaimeyee96@gmail.com (J.Y.); shannon.lxj@gmail.com (S.L.); kellynnoenqixuan@gmail.com (K.O.)

**Keywords:** loneliness, neighbourhood environment, social isolation, socioeconomic status

## Abstract

In Singapore, a densely urbanised Asian city state, more than 80% of the population stays in public housing estates and the majority (90%) own their own homes. For the needy who cannot afford home ownership, public rental flats are available. We were interested in exploring social-environmental factors that are associated with loneliness among elderly residents of public rental housing in Singapore. We surveyed residents aged ≥60 in two Singapore public housing precincts in 2016. Loneliness was measured using a three-item scale. Sociodemographic information was obtained via standardised questionnaires. We used chi-square to identify associations between loneliness and sociodemographic characteristics, as well as neighbourhood perceptions (safety, convenience and the physical environment), on univariate analysis; and logistic regression for multivariate analysis. The response rate was 62.1% (528/800). On multivariate analysis, staying in a rental flat block was independently associated with loneliness (adjusted odds ratio, aOR = 2.10, 95% confidence interval (CI) = 1.32–3.36), as was staying in a poorer physical environment (aOR = 1.92, 95% CI = 1.15–3.22). Although needy Singapore residents share the same built environment as more well-to-do neighbours, differences in the impact of loneliness do exist.

## 1. Introduction

Social relationships are important for health and mental wellness. Loneliness, social isolation, and mental wellness are inextricably interlinked [1,2]. While at the level of the individual, factors such as socioeconomic status (SES), small social networks, and living alone are all associated with loneliness [3,4], the environment that one resides in can also affect loneliness. Evidence suggests that both physical and social characteristics of the neighborhood environment both contribute to loneliness. Residents’ perceived quality of their neighborhood environment is associated with loneliness in studies of deprived neighborhoods in Western societies [5]. Loneliness is a growing problem, particularly in urbanized societies [6]. Some studies have attributed this to changes in the neighborhood environment, brought on by increased urban density and high-rise living. Factors such as transient occupancy and high turnover of residents can reduce familiarity of neighbors in high-rise urban housing, contributing to feelings of isolation [7]. These studies, however, have largely been limited to the context of Western societies. In urbanising Asian societies, loneliness is an increasing problem, driven by shifts in societal norms and living patterns [8,9,10,11,12]. Population ageing and shift towards nuclear families amongst Asian societies may increase the likelihood of social isolation, which is closely tied to loneliness [9]. Various factors of the neighbourhood environment, such as inaccessibility to basic services and lack of maintenance of the physical environment, can contribute to a sense of alienation and thus predispose to loneliness [8]. Neighbourhoods designed to be more walkable can also promote physical activity and social connectedness, with positive effects on health and loneliness; increased mobility can also enhance a sense of environmental mastery and autonomy, improving individual well-being [12]. However, most studies in Asian contexts have explored the links between social characteristics and loneliness; the potential association between perceptions of the neighborhood environment and its impact on loneliness is less well-studied. 

Singapore is one such example of an urbanised multi-ethnic Asian society. Home ownership is a key local indicator of socioeconomic status (SES) in Singapore. The majority of Singaporeans (≥85%) [13,14] stay in public housing and home ownership rates are high (87.2%) [14]. For the needy (<5% of the population) who cannot afford their own home, heavily subsidized public rental housing is available [15]. In Singapore, public rental housing blocks are built within the same neighbourhoods as owner-occupied public housing apartments. Despite the homogeneity of the neighbourhood’s built environment, staying in a public rental flat in Singapore has been correlated with poorer measures of physical and mental health, even after controlling for individual SES (e.g., individual employment status, education, being a recipient of financial aid). Staying in a rental flat neighbourhood was associated with poorer cognitive function [16] and higher depression rates [17] among the elderly. Loneliness amongst residents of public rental housing has not been investigated. In a study of community-dwelling adults in central Singapore, 6.4% reported feelings of loneliness [9]; this increased to 23% in adults aged ≥60 years of age [10]. Loneliness was associated with depressive symptoms [9,18], as well as social isolation [10], with loneliness worsening the psychological effects of living alone [19]. However, the impact of neighbourhood environment on loneliness amongst community-dwelling Singaporeans has not been previously studied. As such, we investigated the prevalence of loneliness and its association with perceptions of the neighbourhood environment and other individual sociodemographic factors, amongst public rental housing residents in Singapore. 

## 2. Methodology

### 2.1. Study Population

We surveyed all residents aged ≥60 years in two public housing precincts in Singapore in September 2016. In Singapore, rental flats are scattered across the island and integrated with owner-occupied blocks in public housing precincts. The public housing precincts chosen in this study were comprised of a mixture of stand-alone rental flat blocks (*n* = 8) and blocks composed solely of owner-occupied housing (*n* = 7). The response rate was calculated based on a combination of census information and information from community grassroots organisations regarding the number of residents aged ≥60 years residing in the blocks. 

### 2.2. Study Methodology

#### 2.2.1. Baseline Information and Measures

At baseline, information on residents’ sociodemographic characteristics, medical, functional and social status was collected via interviewer-administered standardized questionnaires in English, Chinese and Malay. Interviewers were medical students who underwent standardized training prior to study commencement. Comorbidity burden was measured using the Charlson’s Comorbidity Index (CCMI). Functional status in basic activities of daily living (bADL) was also quantified using the Katz Index, while social isolation was quantified using the Lubben’s Social Network Score-6 (LSNS 6) [20]. Health-related quality-of-life (HRQoL) was quantified using the EQ5D [21]. The EQ5D is a standardised measure of health-related quality of life comprising 5 dimensions: mobility, self-care, usual activities, pain/discomfort and anxiety/mood. Each dimension has 5 levels: no problems, slight problems, moderate problems, severe problems and extreme problems. Having mood/anxiety problems was defined as a person indicating that they had at least moderate problems with mood/anxiety on the EQ5D. Additionally, respondents are asked to indicate a global rating of their overall health-related quality of life on a 0–100 point scale. We dichotomised global HRQoL into <75 and ≥75 points, respectively, based on the median values obtained in our study. 

#### 2.2.2. Perceptions of the Neighbourhood Environment

Subjective measures of the neighbourhood environment aimed to assess perceived personal safety, physical convenience and social cohesion within the neighbourhood. We surveyed residents’ perceptions of the neighbourhood environment by using a modified version of the Neighbourhood Environment Walkability Scale-Abbreviated (NEWS-A). The NEWS-A was conceived to provide an empirically derived yet succinct measure of various aspects of the neighbourhood environment. In its original form, the NEWS-A comprised a total of 64 items spread across 12 subscales. However, not all subscales were relevant to the local context. The NEWS has been previously modified for use in the Singaporean setting [22] to study the impact of the neighbourhood environment on physical activity, preserving 8 subscales of residential density, land use mix (diversity), land use mix (access), street connectivity, infrastructure (places for walking and cycling), aesthetics (neighbourhood surroundings), traffic safety and safety from crime. In our study, we utilised the NEWS-A subscales of crime safety, land-use mix (access) and land-use mix (diversity). We omitted the subscales of residential density, aesthetics, infrastructure (walking and cycling), street connectivity, and traffic safety. We removed the neighbourhood density subscale because our study solely focused on public housing estates, which were a homogenous mix of high-rise apartment blocks (no single-storey or detached residences). The street connectivity, traffic safety, aesthetics and infrastructure (walking/cycling) subscales were also removed because these subscales were associated more with physical activity rather than measures of health. In its final form, our study utilised 17 items over 3 subscales (crime safety: 7 items; land use access: 2 items; land use diversity: 8 items) (Table 1). Crime safety and land-use access were originally reported as a 4-point Likert scale; land-use diversity as a 5-point Likert scale. Some of the items were reverse-coded. We summated the responses on the various items to form a total score, as per the scoring system utilised by the NEWS-A. We then used the median modified NEWS-A score to dichotomise the results into “less disadvantaged neighbourhood” and “more disadvantaged neighbourhood”. We further assessed “perceptions of neighborhood safety and convenience” and “perceptions of neighborhood physical environment”; these two factors were derived from factor analysis of the 17 NEWS items utilized in our study (Appendix A). We dichotomised the results for each principal factor, again, using the median result as the cut-off. 

#### 2.2.3. Loneliness

Loneliness was assessed using the three-item UCLA Loneliness Scale [23]. This scale comprises the following three items, assessed on a 3-point scale: “How often do you feel that you lack companionship?”, “How often do you feel left out?” and “How often do you feel isolated from others?”. A score of 6–9 was classified as “lonely” [23]. The UCLA Loneliness Scale has been used previously to measure loneliness amongst community-dwelling adults, in our local population [10].

### 2.3. Statistical Analysis

Descriptive statistics were computed for the study sample. We used chi-square to identify associations between loneliness and sociodemographic characteristics, as well as health-seeking behaviours, on univariate analysis; and backward logistic regression for multivariate analysis to obtain a parsimonious model. All statistical analysis was performed using STATA (Version 22.0, StataCorp, College Station, TX, USA) and statistical significance was set at *p* < 0.05.

### 2.4. Ethics Approval

Ethics approval was obtained from the NUS Institutional Review Board (IRB# B-16-072), informed written consent was obtained, and participation was voluntary.

## 3. Results

A total of 528 residents participated in our study. The response rate was 62.1% (528/800). About half (54.4%, 287/528) of the study population were aged ≥75 years, 70.1% (370/528) had only secondary education and below, and one third (33.9%, 179/528) had a household income of <S$1500/month, compared against the mean household income of S$8,800 in 2016. The median duration of residence in the neighbourhood was 8 years (interquartile ratio, IQR = 5–20). Out of 528 residents, 275 were staying in rental flat blocks, and 253 were staying in owner-occupied housing blocks within the same precincts. About one-third of those staying in rental flat blocks were lonely (32.0%, 88/275), compared with those staying in owner-occupied housing (15.4%, 39/253). About one-third of the study participants (31.6%, 167/528) felt that they lived in a more disadvantaged neighbourhood.

The factors associated with loneliness on univariate analysis are illustrated in Table 1. Overall, on univariate analysis, staying in a stand-alone block, staying in a rental flat, staying in 3-room apartments or smaller, staying in a neighbourhood with greater perceived disadvantage, lower safety and convenience and a poorer perceived physical environment were all factors associated with loneliness, as were being unmarried, not having a religion, being unemployed, having a lower household income, more medical comorbidities, having anxiety/mood issues, poorer functional status, having lower health-related quality of life, and being socially isolated (*p* < 0.05). The factors associated with loneliness on multivariate analysis are illustrated in Table 2. On multivariate analysis, staying in a rental flat apartment was independently associated with loneliness (adjusted odds ratio, aOR = 2.10, 95% confidence interval (CI) = 1.32–3.36), as was staying in a poorer perceived physical environment (aOR = 1.92, 95% CI = 1.15–3.22). Marital status, social isolation, anxiety/mood issues and having a poorer self-rated health-related quality of life were also independently associated with loneliness.

## 4. Discussion

Staying in a rental flat population, as opposed to staying in owner-occupied housing, was associated with loneliness in our study population, even after individual SES and sociodemographic factors were controlled for. While there is a dearth of studies on the prevalence of loneliness amongst rental flat residents in Singapore, a wealth of anecdotal evidence suggests that the problem of loneliness is real [24,25]. Previous estimates of loneliness in the Singaporean population ranged from 6.4% [9] to 23% in adults aged ≥60 years of age [10]. In our population, almost one-third of those staying in rental flats expressed feelings of loneliness. In Singapore, due to a shortage of public rental flats, tenants are required to find a flatmate in order to be eligible for public rental flats; single applicants are provided with a list of single persons who are similarly looking for flatmates [26]. Hence, although tenants of rental flats may not necessarily stay alone, they may not interact much with their flatmate, who may be complete strangers, and hence remain psychologically isolated. Higher turnover of tenants may also contribute to sentiments of insecurity and instability [27]. While all public housing in urbanised Singapore is high-rise, and rental blocks may appear physically indistinguishable on the outside compared to adjacent owner-occupied blocks, staying in a rental flat block may itself be a distinguishing marker that results in stigmatisation and social isolation [27]. Previous studies on public housing in Singapore demonstrated that the frequency of social interaction drops off with increasing physical distance [28]. Hence, the opportunity for interaction between residents of different blocks is limited, even though they may be in the same location.

In our study, poorer perceptions of the neighbourhood physical environment, such as presence of litter, poorer street lighting and signage, and absence of people on the street/common areas were associated with higher odds of loneliness. While the infrastructure in rental and non-rental blocks is similar, anecdotal observations note a preponderance of trash in common areas, as well as a profusion of signages bearing negative messaging (e.g., to beware of illegal moneylenders, or con artists). Immersion “in an environment dominated by negatives” [27] created by a poorly maintained physical environment again reinforces the sense of stigma and distinguishes residents of rental flats from non-rental flats, resulting in increased isolation and loneliness. Furthermore, reduction in attractiveness of common areas serves as a disincentive for residents to linger in the common areas, which is where social and communal interactions take place [28]. Neighbourhood access to amenities and convenience was not independently associated with loneliness in our study, perhaps because of the unique situation in built-up Singapore where both rental and non-rental blocks are co-located, making access to amenities theoretically equal. Given that staying in rental flat blocks in Singapore is significantly associated with loneliness, there may be greater scope for door-to-door services targeted at these enclaves that may increase the potential for re-integration of lonely elderly individuals into the community. In Singapore, various grassroots organizations are involved in activities to reach out to potentially lonely elderly individuals; senior activity centers (SACs) usually found in the vicinity of rental flat neighborhoods can attempt to engage the elderly through community activities and bonding sessions.

The limitations of our study are as follows. While we focused on the associations between loneliness and residents staying in public rental flats, these findings may not be generalizable to those living in other neighbourhoods who may have greater mobility and hence may not be so affected by their immediate living environment. Given the cross-sectional nature of our study, we can only identify correlation, but not causation. Our analysis may also be reflective of an underlying negative response set (i.e., a tendency to report everything negatively, including health and residential factors); however, this is less likely given that we found some factors to be associated with loneliness, but not others.

## 5. Conclusions

In conclusion, although needy Singapore residents share the same built environment as more well-to-do neighbours, differences in loneliness do exist. These differences may be driven by differing perceptions of the physical environment and social environment that reinforce insecurity and stigma.

## Figures and Tables

**Table 1 ijerph-16-00967-t001:** Associations between loneliness and geographical, sociodemographic, medical and social factors, amongst elderly residents in two public housing precincts in Singapore, on univariate analysis (*n* = 528).

Sociodemographic Factors	Not Lonely (*n* = 401) (*n* %)	Lonely (*n* = 127) (*n* %)	OR (95% CI)	*p*-Value
Geographical				
Stayed in neighbourhood for >8 years				
No	241 (78.8)	65 (21.2)	1.00	0.080
Yes	160 (72.1)	62 (27.9)	1.44 (0.96–2.15)	
Staying in rental apartment vs. owner-occupied				
Owner-occupied	213 (84.5)	39 (15.5)	1.00	<0.001
Rental	188 (68.1)	88 (31.9)	2.56 (1.67–3.91)	
Number of rooms				
3 rooms or smaller	263 (71.7)	104 (28.3)	1.00	<0.001
4–5 rooms	138 (85.7)	23 (14.3)	0.42 (0.26–0.69)	
Perceived neighbourhood disadvantage				
Less disadvantaged	288 (79.8)	73 (20.2)	1.00	0.003
More disadvantaged	113 (67.7)	54 (32.3)	1.89 (1.25–2.85)	
Perceptions of neighbourhood safety and convenience				
Less safe and convenient	175 (70.6)	73 (29.4)	1.00	0.008
More safe and convenient	226 (80.7)	54 (19.3)	0.57 (0.38–0.86)	
Perceptions of neighbourhood physical environment				
Better physical environment	269 (73.5)	97 (26.5)	1.00	0.049
Poorer physical environment		30 (18.5)	0.63 (0.40–0.98)	
Socio-demographic				
Gender				
Female	235 (76.3)	73 (23.7)	1.00	0.837
Male	166 (75.5)	54 (24.5)	1.05 (0.70–1.57)	
Marital status				
Not married	168 (70.0)	72 (30.0)	1.00	0.004
Married	233 (80.9)	55 (19.1)	0.55 (0.37–0.82)	
Religious				
No	113 (68.1)	53 (31.9)	1.00	0.006
Yes	288 (79.6)	74 (20.4)	0.55 (0.36–0.83)	
Age				
Age 60–75 years	193 (80.1)	48 (19.9)	1.00	0.052
Age ≥ 75 years	208 (72.5)	79 (27.5)	1.53 (1.01–2.30)	
Currently employed				
No	207 (71.4)	83 (28.6)	1.00	0.008
Yes	194 (81.5)	44 (18.5)	0.57 (0.37–0.86)	
Education				
Secondary and below	121 (76.6)	37 (23.4)	1.00	0.912
Post-secondary and above	280 (75.7)	90 (24.3)	1.05 (0.68–1.63)	
Number of people in household				
2 or less people	232 (79.2)	61 (20.8)	1.00	0.065
3 or more people	169 (71.9)	66 (28.1)	1.49 (1.00–2.22)	
Average household income				
≤$1500/month	250 (71.6)	99 (28.4)	1.00	0.001
>$1500/month	151 (84.4)	28 (15.6)	0.47 (0.29–0.75)	
Medical and functional status				
Comorbidity (Charlson Comorbidity Index)				
CCMI = 0	308 (78.2)	86 (21.8)	1.00	0.047
CCMI ≥ 1	93 (69.4)	41 (30.6)	1.58 (1.02–2.45)	
Chronic pain (pain > 6 months)				
No	359 (76.7)	109 (23.3)	1.00	0.263
Yes	42 (70.0)	18 (30.0)	1.41 (0.78–2.55)	
Anxiety/mood issues				
No	386 (78.8)	104 (21.2)	1.00	<0.001
Yes	15 (39.5)	23 (60.5)	5.69 (2.86–11.30)	
Functional status (basic activities of daily living)				
Dependent in at least 1 bADL	8 (47.1)	9 (52.9)	1.00	0.008
Independent in all bADLs	393 (76.9)	118 (23.1)	0.27 (0.10–0.71)	
State of perfect health (EQ5D)				
No	135 (66.5)	68 (33.5)	1.00	<0.001
Yes	266 (81.8)	59 (18.2)	0.44(0.29–0.66)	
Global health-related quality of life				
No	194 (69.8)	84 (30.2)	1.00	
Yes	207 (82.8)	43 (17.2)	0.48 (0.32–0.73)	0.001
Social network				
Has caregiver				
No	323 (76.9)	97 (23.1)	1.00	0.315
Yes	78 (72.2)	30 (27.8)	1.28 (0.79–2.07)	
Social isolation (Lubbens Social Network Score-6)				
No (LSNS < 12)	188 (88.3)	25 (11.7)	1.00	<0.001
Yes (LSNS ≥ 12)	213 (67.6)	102 (32.4)	3.60 (2.23–5.82)	

CCMI: Charlson’s Comorbidity Index; bADL: basic activities of daily living; EQ5D: a standardised measure of health-related quality of life; LSNS: Lubbens Social Network Score.

**Table 2 ijerph-16-00967-t002:** Associations between loneliness and geographical, sociodemographic, medical and social factors, amongst elderly residents in two public housing precincts in Singapore, on multivariate analysis (*n* = 528).

Increased Loneliness	aOR (95% CI)	*p*-Value
Staying in rental apartment vs. owner-occupied		
Owner-occupied	1.00	0.002
Rental apartment	2.10 (1.32–3.36)	
Perceptions of neighbourhood physical environment		
Better physical environment	1.00	0.012
Poorer physical environment	1.92 (1.15–3.22)	
Social isolation (Lubbens Social Network Score-6)		
No (LSNS < 12)	1.00	<0.001
Yes (LSNS ≥ 12)	3.25 (1.96–5.40)	
Marital status		
Not married	1.00	0.027
Married	0.60 (0.39–0.95)	
Anxiety/mood issues		
No	1.00	<0.001
Yes	5.68 (2.1–11.88)	
Global health-related quality of life		
No	1.00	0.003
Yes	0.51 (0.33–0.80)	

aOR: Adjusted Odds Ratio; CI: confidence interval.

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
