# Peer review of "Loneliness amongst Low-Socioeconomic Status Elderly Singaporeans and its Association with Perceptions of the Neighbourhood Environment"

_ijerph, 2019, doi:10.3390/ijerph16060967_

Round 1

Reviewer 1 Report

Your work is important to the health of the elderly in Singapore. I hope your government takes notice and action based on your extensive results. I have the following suggestions.

Consistency in referencing is needed (reference). Then at time it is .(reference)

Line 72, I would spell out September.

Line 96, ....environment aimed...

Line 119-130, It seems you are reporting results in the methods section. I would report all the information in the results section.

Table 1, Anxiety/mood, those are some very wide confidence intervals. At the upper end, one is really at risk for loneliness with the anxiety/mood issue OR 95% UL = 11.30.

Author Response

Reply to Reviewer 1:

Your work is important to the health of the elderly in Singapore. I hope your government takes notice and action based on your extensive results.

We thank the reviewer for the kind comments.

 I have the following suggestions

Consistency in referencing is needed (reference). Then at time it is .(reference)

We have standardized the referencing accordingly.

Line 72, I would spell out September.

This has been done.

Line 96, ....environment aimed...

This has been done.

Line 119-130, It seems you are reporting results in the methods section. I would report all the information in the results section.

We thank the reviewers for the feedback.

The line on “About one-third of the study participants (31.6%, 167/528) felt that they lived in a more disadvantaged neighbourhood.” has been moved to the Results section.

We have revised the following lines to read, “We further assessed “perceptions of neighborhood safety and convenience” and “perceptions of neighborhood physical environment”; these two factors were derived from factor analysis of the 17 NEWS items utilized in our study (Supplementary Table 1). We dichotomised the results for each principal factor, again, using the median result as the cut-off”

The following lines, “Factor analysis of the 17 NEWS items produced 2 principal components, summarised as “perceptions of neighbourhood safety and convenience” and “perceptions of neighbourhood physical environment”. Most of the subscales related to perceived neighbourhood safety (eg. perceived crime rate in neighbourhood, perceived safety when walking around) and perceived neighbourhood convenience (availability of amenities, such as convenient access to shops, eating places, medical services, and recreational areas) loaded onto the first principal component, while subscales related to the physical environment (eg. presence of litter, lighting/signage, physical barriers such as uneven ground or steps) loaded onto the second principal component.” have been moved as a footnote to Supplementary Table 1.

Table 1, Anxiety/mood, those are some very wide confidence intervals. At the upper end, one is really at risk for loneliness with the anxiety/mood issue OR 95% UL = 11.30.

Yes, there was a strong association noted between issues with anxiety/mood reported on the EQ5D (self-reported measure of health-related quality of life) and loneliness in our study population.

Reviewer 2 Report

There is no question that loneliness among older adults is a concern, but the question of "so what?" comes to mind with these results. Why is it important to study the prevalence of loneliness and its association with perceptions of the neighborhood environment? Perhaps the introduction could better discuss the context of urbanising Asian societies and review the literature that helps the authors to hypothesize why older adults living in public rental housing might have higher rates of loneliness. Is this because of demographic shifts? changing familial expectations? poorer individuals having different profiles? Please discuss the reliability and validity of these western measures for this population.

Author Response

There is no question that loneliness among older adults is a concern, but the question of "so what?" comes to mind with these results. Why is it important to study the prevalence of loneliness and its association with perceptions of the neighborhood environment? Perhaps the introduction could better discuss the context of urbanising Asian societies and review the literature that helps the authors to hypothesize why older adults living in public rental housing might have higher rates of loneliness. Is this because of demographic shifts? changing familial expectations? poorer individuals having different profiles?

We thank the Reviewer for the feedback. We have added the following sentences to the Introduction,

“ In urbanising Asian societies, loneliness is an increasing problem, driven by shifts in societal norms and living patterns.(8-12) Population ageing and shift towards nuclear families amongst Asian societies may increase the likelihood of social isolation, which is closely tied to loneliness.(9)Various factors of the neighbourhood environment, such as inaccessibility to basic services and lack of maintenance of the physical environment, can contribute to a sense of alienation and thus predispose to loneliness.(8) Neighbourhoods designed to be more walkable can also promote physical activity and social connectedness, with positive effects on health and loneliness; increased mobility can also enhance a sense of environmental mastery and autonomy, improving individual well-being.(12) However, most studies in Asian contexts have explored the links between social characteristics and loneliness; the potential association between perceptions of the neighborhood environment and its impact on loneliness is less well-studied.  ”

Please discuss the reliability and validity of these western measures for this population.

We thank the reviewer for the feedback.

In our Methodology, we highlight that the primary outcome measure of loneliness was measured using the UCLA Loneliness Scale, which has previously been validated in our local population. We have rephrased the following lines to read:

“The UCLA Loneliness Scale has been used previously to measure loneliness amongst community-dwelling adults, in our local population.(10)”